# More precise method of low-density lipoprotein cholesterol estimation for tobacco and electronic cigarette smokers: A cross-sectional study

**Han-Joon Bae** * , Hae Won Jung, Seung-Pyo Hong

Division of Cardiology, Department of Internal Medicine, Daegu Catholic University Medical Center, Daegu, South Korea

* dcmcbae@cu.ac.kr

## Abstract

Smoking is associated with elevated low-density lipoprotein cholesterol (LDL-C) levels. However, the accuracies of the Friedewald, Sampson, and Martin LDL-C-estimating equations based on smoking status are unclear. We analyzed the accuracy of LDL-C levels estimated using these three equations based on tobacco and electronic cigarette smoking status. Data on LDL-C and other lipid components were obtained from the Korea National Health and Nutrition Examination Survey from January 2009 to December 2021. Direct LDL-C (dLDL-C) levels and smoking data of 12,325 participants were evaluated. Current smokers had higher triglyceride levels than never smokers. Electronic cigarette smokers had higher triglyceride and dLDL-C levels than never smokers. The Martin equation yielded more accurate mean absolute deviations than the other equations for the group with triglyceride levels <400 mg/dL as well as more accurate median absolute deviation values, except for the group with dLDL-C levels <40 mg/dL. Similar estimates were derived from the equations when the triglyceride levels were <150 mg/dL. However, the Martin equation may lead to the overestimation of LDL-C levels. In conclusion, the Martin equation is suitable for triglyceride levels <400 mg/dL regardless of the electronic cigarette/tobacco smoking status; if the triglyceride level is <150 mg, the Friedewald equation could also be considered, regardless of the electronic cigarette/tobacco smoking status.

## Introduction

Cardiovascular disease (CVD) is the predominant cause of death and is associated with an increasing number of diseases worldwide [1]. Low-density lipoprotein cholesterol (LDL-C) is a major risk factor for CVD and is the primary target for preventing atherosclerotic cardiovascular events [2–5]. Additionally, the clinical significance of lipid modification is a key factor in atherosclerotic CVD.

**Funding:** This work was supported by a grant from the Research Institute of Medical Science, Daegu Catholic University, Daegu, South Korea (2022, RD-22-0007). The funders had no role in considering the study design or in the collection, analysis, interpretation of data, writing of the report, or decision to submit the article for publication.

**Competing interests:** NO authors have competing interests

Smoking is the most common form of tobacco use worldwide. It is a well-known, significant classical risk factor for CVD and one of the primary causes of death globally [6, 7]. Cigarette smoke is a complex, dynamic chemical mixture composed of multiple compounds. Specifically, it contains more than 4,000 chemicals as well as numerous compounds of variable sizes that constitute particulate matter [6–8]. All forms of cigarette smoke are harmful, and there is no safe level of exposure to it. Additionally, smoke exposure can cause atherogenesis, atherosclerosis, and atherothrombosis [8], and both smoking and smoking cessation have been shown to be associated with changes in cholesterol levels [9, 10].

The best method for measuring LDL-C levels uses ultracentrifuged plasma, but this method is time-consuming and labor-intensive [11]. Therefore, several homogeneous direct methods to measure high-density lipoprotein cholesterol (HDL-C) and LDL-C levels have been developed. These methods use various surfactants, ionic polymers, and other components to measure cholesterol levels in specific classes of lipoproteins in the serum [12]; however, they can be expensive and are not available at all medical centers. Currently, LDL-C is typically calculated using three equations: the Friedewald, Sampson, and Martin equations [13, 14]. However, it is unclear which of these can estimate LDL-C levels in patients more accurately based on their smoking status. Moreover, the use of electronic cigarettes (ECs) has increased considerably recently, but there have been very few studies on LDL-C measurements in this context.

In this study, we examined the accuracy of LDL-C estimation using the three equations based on tobacco cigarette (TC) and EC smoking status.

## Materials and methods

### Study design and participants

This cross-sectional study utilized data obtained from the Korea National Health and Nutrition Examination Survey (KNHANES) [15] of the South Korean population. Detailed information can be found in a previous study [16]. This study was conducted in accordance with the Declaration of Helsinki and was approved by the institutional review boards of each location (IRB 2018-01-03-P-A, 2018-01-03-2C-A). In the KNHANES, a cohort of 20,388 South Korean adults aged ≥20 years was included between January 2009 and December 2021. Participants with available estimated LDL-C and smoking questionnaire data were included in this study. Individuals with triglyceride (TG) levels >1000 mg/dL and missing cholesterol data were excluded. The detailed inclusion and exclusion criteria are illustrated in S1 Fig.

### Lipid sample measurement and lipid level estimation

Blood samples were obtained after a minimum fasting period of 8 h. The samples were analyzed by a homogeneous direct assay using reagents from Sekisui Medical Corporation (Tokyo, Japan) on a Hitachi 7600 automated analyzer (Hitachi, Tokyo, Japan), as previously described [16].

### Smoking status

The participants were categorized into three groups according to their smoking status. Current smokers were defined as adults who were actively engaged in cigarette smoking and had a history of smoking cigarettes throughout their lifetime. Former smokers were defined as adults who had a history of smoking >100 cigarettes in their lifetime but had quit smoking at the time of the interview. Never smokers were defined as adults who had either never smoked cigarettes (TC and EC) or had smoked <100 cigarettes throughout their lifetime. EC smokers

were defined as adults who were currently engaged in vaping. They shared the defined characteristics of former and current smokers but not of never smokers.

## Statistical analyses

Continuous variables are presented as medians and interquartile ranges [IQRs], while categorical variables are presented as numbers and percentages (%). Statistical analysis was conducted by smoking status for each TG level. TG levels were classified into three categories: <150, 150–400, and 400–1000 mg/dL. Overall precision was defined as the ratio of direct LDL-C (dLDL-C) to estimated LDL-C and non-high-density lipoprotein (NHDL) (S2–S4 Figs).

The correlations among the three equations and dLDL-C levels were represented through scatter plots, stratified by each smoking status. Concordance was evaluated using statistical metrics, including mean absolute error (MAE), $R^2$, and root mean square error (RMSE). Residual error was defined as an estimate of the disparity between dLDL-C values and estimated LDL-C values. The mean absolute difference (MAD) and MAE values were computed, and confidence intervals (CIs) for median absolute deviations (MeADs) were established for each equation difference, stratified according to dLDL-C levels. The R package DescTools was used to calculate the CIs for the MeADs of two-sample difference and deviation of the sample. All statistical analyses were executed using R version 4.1.3 (R Foundation for Statistical Computing, Vienna, Austria).

## Results

### Characteristics of LDL-C stratified by TG levels

In the KNHANES, the dLDL-C levels of 12,325 South Korean individuals (median age, 51 [38–63] years; 54.8% male) were determined. The demographic characteristics and lipid profile of the participants stratified by TG levels and smoking status are shown in S1–S3 Tables.

At each TG level, the current smoker group exhibited higher TG levels (except those with TG levels <150 mg/dL) and dLDL-C levels (except those with TG levels <150 mg/dL) than the never smoker group (except those with TG levels <150 mg/dL). The EC smoker group had higher TG, lower HDL, and higher dLDL-C levels than the never smoker group. No significant differences were observed among the three equations (Friedewald, Sampson, and Martin) for those with TG levels <150 mg/dL. The current smoker group exhibited a lower calculated LDL-C value than the never smoker group when TG levels were ≥150 mg/dL. The positive absolute value of calculated LDL-C was higher in the current smoker group than in the never smoker group. Additionally, the absolute value of LDL-C for the never smoker group with TG levels ≥150 mg/dL determined using the Sampson equation was lower than the values determined using other equations (Friedewald and Martin).

### Concordance of direct and estimated LDL-C

The correlations between the estimated LDL-C values and dLDL-C levels, stratified by smoking status, are illustrated in S2 Fig. In the never smoker group, the correlation coefficients were 0.826 (Sampson), 0.818 (Martin), and 0.796 (Friedewald); the corresponding MAEs were 7.06, 7.27, and 8.85. In the former smoker group, the correlation coefficients were 0.919 (Sampson), 0.814 (Martin), and 0.790 (Friedewald), and the corresponding MAEs were 7.55, 7.45, and 11.35. In the current smoker group, the correlation coefficients were 0.811 (Sampson), 0.815 (Martin), and 0.773 (Friedewald); the corresponding MAEs were 8.19, 7.69, and 11.60, respectively. In the EC smoker group, the correlation coefficients were 0.795 (Sampson), 0.797 (Martin), and 0.754 (Friedewald); the corresponding MAEs were 8.40, 7.95 and 11.37.

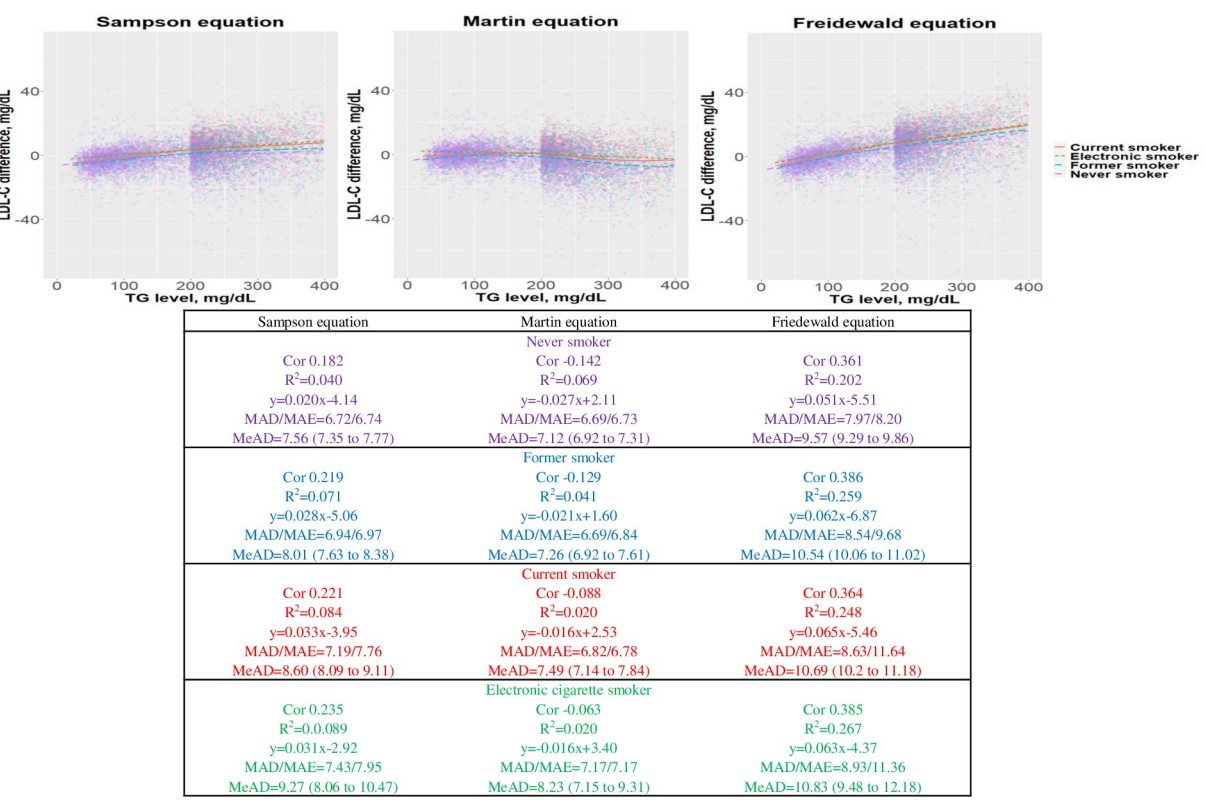

**Fig 1. Residual error plots for low-density lipoprotein cholesterol (LDL-C) level derived by equations according to smoking status (triglyceride level <400 mg/dL).** Differences between the estimates obtained by the Sampson equation (A), Martin equation (B), and Friedewald equation (C) and dLDL-C, stratified by TG level. Residual error was calculated as the difference between dLDL-C and estimated LDL-C values. The dots indicate the individual samples, colored according to smoking status. The color scale indicates individuals. The solid line indicates the trend by the local regression method. MeADs with 95% CIs were calculated by two-sample difference. Cor, correlation coefficient; MAD, mean absolute deviation; MAE, mean absolute error; Median absolute deviation, MeAD; $R^2$, correlation coefficient.

## Residual error and MAD

The correlations between differences in the estimated LDL-C and dLDL-C values and TG levels, according to smoking status, are illustrated in Fig 1 (TG levels <400 mg/dL) and S5 Fig (TG levels <1000 mg/dL). In the never smoker group, the coefficients of correlation with TG levels <400 mg/dL were 0.182 (Sampson), -0.142 (Martin), and 0.361 (Friedewald); the corresponding MADs were 6.72, 6.69, and 7.97; and the MeADs with 95% CIs were 7.56 (7.35–7.77), 7.12 (6.92–7.31), and 9.57 (9.29–9.86). In the former smoker group, the coefficients of correlation with TG levels <400 mg/dL were 0.219 (Sampson), -0.129 (Martin), and 0.386 (Friedewald); the corresponding MADs were 6.94, 6.69, and 8.54. In the current smoker group, the coefficients of correlation between LDL-C and TG levels <400 mg/dL were 0.221 (Sampson), -0.088 (Martin), and 0.364 (Friedewald); the corresponding MADs were 7.19, 6.82, and 8.63. In the EC smoker group, the coefficients of the correlation with TG levels <400 mg/dL were 0.235 (Sampson), -0.063 (Martin), and 0.385 (Friedewald); the corresponding MADs were 7.43, 7.17, and 8.93.

## Mean absolute differences stratified by lipid components

The MADs of the three equations according to the dLDL-C levels classified by smoking status are illustrated in Fig 2 (TG levels <150 mg/dL and TG levels <1000 mg/dL). In the never

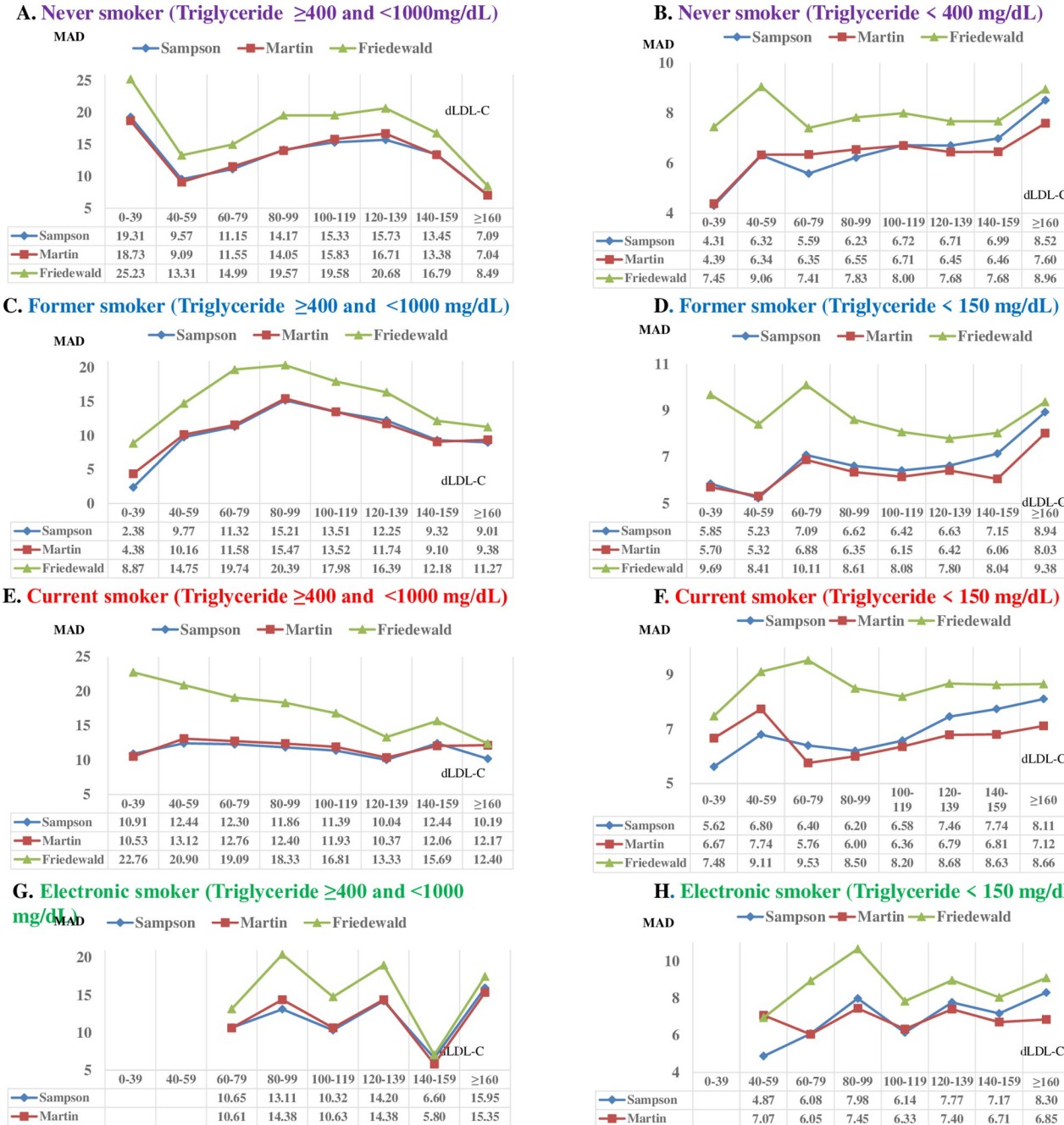

**Fig 2. Mean absolute deviations of directly measured low-density lipoprotein cholesterol (dLDL-C) stratified by estimated LDL-C.** (A) MAD of estimated LDL-C vs. dLDL-C for the never smoker group (TG levels ≥400 and <1000 mg/dL [4]). (B) MAD of estimated LDL-C vs. dLDL-C for the never smoker group (TG levels <400 mg/dL). (C) MAD of estimated LDL-C vs. dLDL-C for the former smoker group (TG levels TG levels ≥400 and <1000 mg/dL). (D) MAD of estimated LDL-C vs. dLDL-C for the former smoker group (TG levels <400 mg/dL). (E) MAD of estimated LDL-C vs. dLDL-C for the current smoker group (TG levels TG levels ≥400 and <1000mg/dL). (F) MAD of estimated LDL-C vs. dLDL-C for the current smoker group (TG levels <400 mg/dL). (G) MAD of estimated LDL-C vs. dLDL-C for the EC smoker group (TG levels TG levels ≥400 and <1000 mg/dL). (H) MAD of estimated LDL-C vs. dLDL-C for the EC smoker group (TG levels <400 mg/dL). HDL, high-density lipoprotein; TG, triglyceride; EC, electronic cigarette. Sampson equation (blue), Martin equation (red), and Friedewald equation (green).

smoker group with TG levels <150 mg/dL, MADs were 8.84, 8.45, and 8.97 for dLDL-C levels <40 mg/dL; 4.91, 4.71, and 5.12 for dLDL-C levels ≥40 and <59 mg/dL; and 4.17, 3.89, and 4.45 for dLDL-C levels ≥60 and <80 mg/dL. In the former smoker group with TG levels <150 mg/dL, MADs were 10.76, 10.20, and 10.69 for dLDL-C levels <40 mg/dL; 5.28, 5.10, and 5.33 for dLDL-C levels ≥40 and <60 mg/dL; and 4.05, 3.78, and 4.31 for dLDL-C levels ≥60 and <80 mg/dL.

In the current smoker group with TG levels <150 mg/dL, the MADs were 5.82, 5.11, and 6.62 for dLDL-C levels <40 mg/dL; 4.14, 3.99, and 4.60 for dLDL-C levels ≥40 and <60 mg/dL; and 4.15, 3.88, and 4.61 for dLDL-C levels ≥60 and <80 mg/dL. In the EC smoker group with TG levels <150 mg/dL, the MADs were 3.00, 3.13, and 3.82 for dLDL-C levels <40 mg/dL; 4.19, 3.99, and 4.73 for dLDL-C levels ≥40 and <60 mg/dL; and 4.54, 4.22, and 4.82 for dLDL-C levels ≥60 and <80 mg/dL.

## Precision of estimated LDL-C stratified by LDL-C

MADs and MeADs with 95% CIs are shown in Table 1 (TG levels <150 mg/dL), S4 Table (TG levels <400 mg/dL), and S5 Table (TG levels <1000 mg/dL).

In the never smoker group with TG levels <150 mg/dL and dLDL-C levels <70 mg/dL, MADs and MeADs were as follows: Sampson equation, 4.23 and -1.63 (-3.35–0.09) vs. Martin equation, 4.07 and -1.48 (-3.18–0.21) vs. Friedewald equation, 4.47 and -1.33 (-3.01–0.34). In the former smoker group with TG levels <150 mg/dL and dLDL-C levels <70 mg/dL, MADs and MeADs were as follows: Sampson equation, 3.90 and -1.93 (-8.10–4.25) vs. Martin equation, 3.44 and -1.33 (-7.46–4.80) vs. Friedewald equation, 4.71 and -4.27 (-10.85–2.31).

In the current smoker group with TG levels <150 mg/dL and dLDL-C levels <70 mg/dL, MADs and MeADs were as follows: Sampson equation, 3.89 and -3.71 (-7.77–0.35) vs. Martin equation, 3.73 and -3.26 (-7.21–0.69) vs. Friedewald equation, 4.41 and -4.11 (-8.25–0.04). In the EC smoker group with TG levels <150 mg/dL and dLDL-C levels <70 mg/dL, MADs and MeADs were as follows: Sampson equation, 4.65 and -7.41 (-15.16–0.33) vs. Martin equation, 4.56 and -8.01 (-22.39–6.38) vs. Friedewald equation, 5.57 and -6.69 (-14.09–0.71).

The Friedewald equation demonstrated higher MADs and MeADs than the other equations for TG levels <150 mg/dL, <400 mg/dL, 400–1000 mg/dL, and <1000 mg/dL according to smoking status (Table 1 and S4–S6 Tables). The Martin equation for TG levels <150 mg/dL and 150–400 mg/dL demonstrated lower MADs and MeADs than the other equations. The Sampson equation demonstrated lower MADs and MeADs for TG levels 400–1000 mg/dL and <400 mg/dL and dLDL-C levels <100 mg/dL. In case of LDL-C levels <70 mg/dL and TG levels >400 mg/dL, the MAD of the Sampson equation was lower than those of the other equations without never smokers.

## Precision of estimated LDL-C for the group with daily smoker status

The MADs and MeADs with 95% CIs for the current TC and EC smoking status are illustrated in Figs 3 and 4. In the current smoker group with TG levels <400 mg/dL, the MADs, MeADs, and correlation coefficients obtained for the three equations were similar. However, as the TG level increased, the differences in MADs and correlation coefficients also increased for each equation. The accuracies of the Martin and Sampson equations were similar to that of the Friedewald equation. However, in the current smoker group with TG levels >400 mg/dL, the Martin equation exhibited a slight overestimation. The Friedewald equation exhibited a higher difference than the other equations. However, for TG levels <150 mg/dL, all three equations showed acceptable ranges according to smoking burden.

**Table 1. Mean and median absolute deviations with 95% CIs of estimated low-density lipoprotein cholesterol stratified by dLDL-C in the group with TG levels of <150 mg/dL.**

| | Sampson equation | | Martin equation | | Friedewald equation | |
|---|---|---|---|---|---|---|
| | | | *Never smoker* | | | |
| **dLDL-C** | **MAD** | **MeAD** | **MAD** | **MeAD** | **MAD** | **MeAD** |
| <40 | 3.99 | -0.96 (-5.85–3.93) | 3.75 | -2.08 (-6.68–2.53) | 4.40 | -1.60 (-5.65–2.45) |
| <70 | 4.23 | -1.63 (-3.35–0.09) | 4.07 | -1.48 (-3.18–0.21) | 4.47 | -1.33 (-3.01–0.34) |
| ≥70 and <100 | 4.31 | -0.89 (-1.76 to -0.02) | 3.98 | 0.00 (-0.83–0.83) | 4.54 | -0.64 (-1.50–0.23) |
| ≥100 and <130 | 4.92 | -1.85 (-2.72 to -0.98) | 4.47 | -1.48 (-2.34 to -0.63) | 5.09 | -1.10 (-1.93 to -0.27) |
| ≥130 and <160 | 5.24 | -1.33 (-2.57 to -0.10) | 4.83 | -0.74 (-1.94–0.46) | 5.27 | -1.02 (-2.24–0.21) |
| ≥160 | 6.65 | 0.00 (-5.55–5.55) | 6.46 | 0.52 (-4.99–6.03) | 6.64 | 0.16 (-5.38–5.71) |
| | | | *Former smoker* | | | |
| **dLDL-C** | MAD | MeAD | MAD | MeAD | MAD | MeAD |
| <40 | 3.95 | -6.23 (-22.96–10.51) | 2.73 | -3.85 (-15.32–7.61) | 5.44 | -5.45 (-19.62–8.72) |
| <70 | 3.90 | -1.93 (-8.10–4.25) | 3.44 | -1.33 (-7.46–4.80) | 4.71 | -4.27 (-10.85–2.31) |
| ≥70 and <100 | 4.10 | 0.74 (-1.09–2.57) | 3.72 | 0.89 (-0.94–2.72) | 4.58 | 1.07 (-0.73–2.88) |
| ≥100 and <130 | 4.65 | -0.74 (-2.51–1.03) | 4.37 | -0.59 (-2.36–1.17) | 4.82 | -0.14 (-1.83–1.55) |
| ≥130 and <160 | 4.49 | -0.44 (-2.89–2.00) | 4.27 | 0.22 (-2.16–2.61) | 4.53 | -0.56 (-3.00–1.89) |
| ≥160 | 7.97 | -1.85 (-11.55–7.84) | 7.62 | -0.89 (-10.47–8.69) | 7.97 | -1.76 (-11.44–7.92) |
| | | | *Current smoker* | | | |
| **dLDL-C** | MAD | MeAD | MAD | MeAD | MAD | MeAD |
| <40 | 1.00 | -1.48 (-27.64–24.68) | 1.55 | -2.30 (-30.81–26.22) | 1.39 | -2.05 (-29.85–25.74) |
| <70 | 3.89 | -3.71 (-7.77–0.35) | 3.73 | -3.26 (-7.21–0.69) | 4.41 | -4.11 (-8.25–0.04) |
| ≥70 and <100 | 3.64 | -0.89 (-4.27–2.49) | 3.31 | -0.44 (-3.78–2.90) | 3.96 | -1.21 (-4.63–2.21) |
| ≥100 and <130 | 4.33 | -2.08 (-4.23–0.08) | 3.83 | -1.04 (-3.09–1.01) | 4.64 | -1.90 (-4.05–0.25) |
| ≥130 and <160 | 5.72 | -1.93 (-7.18–3.33) | 5.07 | -0.89 (-6.20–4.42) | 5.85 | -2.28 (-7.82–3.25) |
| ≥160 | 8.25 | -6.30 (-14.82–2.21) | 7.85 | -5.26 (-13.37–2.84) | 8.14 | -5.78 (-14.01–2.45) |
| | | | *Electronic cigarette smoker* | | | |
| **dLDL-C** | **MAD** | **MeAD** | **MAD** | **MeAD** | **MAD** | **MeAD** |
| *<40* | | | | | | |
| <70 | 4.65 | -7.41 (-15.16–0.33) | *4.56* | -8.01 (-22.39–6.38) | 5.57 | -6.69 (-14.09–0.71) |
| ≥70 and <100 | 4.52 | -0.74 (-5.30–3.82) | *4.20* | -0.30 (-5.10–4.51) | 4.69 | -2.42 (-7.17–2.33) |
| ≥100 and <130 | 3.85 | 0.22 (-3.*26*–3.71) | *3.79* | *0.74 (-2.89–4.37)* | 3.96 | -0.34 (-3.83–3.15) |
| ≥130 and <160 | 4.02 | 3.56 (-4.*20*–11.32) | 3.42 | 2.37 (-5.32–10.06) | 4.23 | 3.65 (-3.92–11.23) |
| ≥160 | 7.35 | 0.22 (-28.27–28.71) | 6.97 | 1.19 (-28.04–30.42) | 7.43 | 0.11 (-28.7–28.92) |

CI, confidence interval; dLDL-C, direct low-density lipoprotein cholesterol; MAD, mean absolute deviation; MeAD, median absolute deviation. The MeADs with 95% CIs were calculated by two-sample difference.

SI conversion factors: To convert cholesterol to mmol/L, values were multiplied by 0.0259.

In the EC smoker group with TG levels <400 mg/dL and >400 mg/dL, the MADs, MeADs, and correlation coefficients for the Sampson and Martin equations were similar. The Friedewald equation exhibited a higher difference than the other equations.

## Discussion

In this investigation, we evaluated and compared the precision of LDL-C levels determined by three equations according to smoking status. This study revealed four main findings: 1) current smokers exhibited a higher TG level than never smokers; 2) the Sampson equation exhibited superior precision to other equations for the group with TG levels >400 mg/dL, regardless

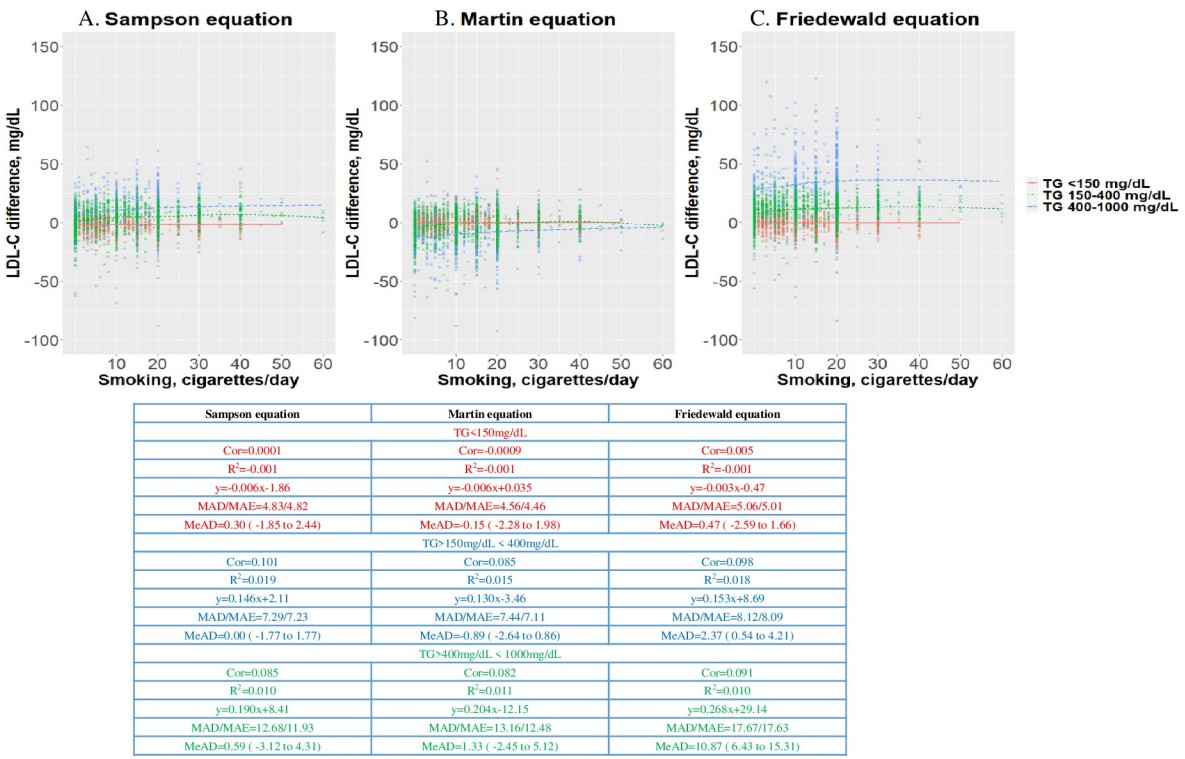

| Sampson equation | Martin equation | Friedewald equation |
|---|---|---|
| **TG<150mg/dL** | | |
| Cor=0.0001 | Cor=-0.0009 | Cor=0.005 |
| R²=-0.001 | R²=-0.001 | R²=-0.001 |
| y=-0.006x-1.86 | y=-0.006x+0.035 | y=-0.003x-0.47 |
| MAD/MAE=4.83/4.82 | MAD/MAE=4.56/4.46 | MAD/MAE=5.06/5.01 |
| MeAD=0.30 ( -1.85 to 2.44) | MeAD=-0.15 ( -2.28 to 1.98) | MeAD=0.47 ( -2.59 to 1.66) |
| **TG>150mg/dL < 400mg/dL** | | |
| Cor=0.101 | Cor=0.085 | Cor=0.098 |
| R²=0.019 | R²=0.015 | R²=0.018 |
| y=0.146x+2.11 | y=0.130x-3.46 | y=0.153x+8.69 |
| MAD/MAE=7.29/7.23 | MAD/MAE=7.44/7.11 | MAD/MAE=8.12/8.09 |
| MeAD=0.00 ( -1.77 to 1.77) | MeAD=0.89 ( -2.64 to 0.86) | MeAD=2.37 ( 0.54 to 4.21) |
| **TG>400mg/dL < 1000mg/dL** | | |
| Cor=0.085 | Cor=0.082 | Cor=0.091 |
| R²=0.010 | R²=0.011 | R²=0.010 |
| y=0.190x+8.41 | y=0.204x-12.15 | y=0.268x+29.14 |
| MAD/MAE=12.68/11.93 | MAD/MAE=13.16/12.48 | MAD/MAE=17.67/17.63 |
| MeAD=0.59 ( -3.12 to 4.31) | MeAD=1.33 ( -2.45 to 5.12) | MeAD=10.87 ( 6.43 to 15.31) |

**Fig 3. Distribution plots showing low-density lipoprotein cholesterol (LDL-C) levels measured by different equations according to cigarettes per day.** Differences between the estimates obtained by the Sampson equation (A), Martin equation (B), and Friedewald equation and direct LDL-C (dLDL-C), stratified by cigarettes per day. The dots indicate the individual samples, colored according to TG levels. The color scale indicates individuals. The solid line indicates the trend by the local regression method. Cor, correlation coefficient; MAD, mean absolute deviation; MAE, mean absolute error; MeAD, median absolute deviation; R², correlation coefficient; RMSE, root mean square error; TG, triglyceride.

of the smoking status; 3) the Martin equation exhibited greater precision than the other equations for the group with TG levels <400 mg/dL, irrespective of the smoking status; and 4) the Friedewald equation was not more accurate than the other equations. However, the differences in the MADs, MeADs, and residual errors with and without EC and TC smoking status was not substantial for TG levels <150 mg/dL with all three equations.

## Relationship between lipoprotein and smoking

Smoking decreases HDL-C levels, while smoking cessation is linked to increased HDL-C levels as measured in older adults [9, 10]; however, this association is not completely understood. Smoking induces the release of catecholamines, leading to an increase in circulating fatty acids; this process may lead to elevated concentrations of LDL and very low-density lipoprotein (VLDL) as well as decreased concentrations of HDL-C [17]. Smoking additionally decreases the levels of lecithin cholesterol acyltransferase, an enzyme responsible for esterifying free cholesterol and increasing HDL size. Smoking cessation was associated with reductions in TG levels, potentially attributed to the counterbalancing effect of weight gain; nevertheless, no significant changes were observed in LDL-C level, LDL particle concentration, or LDL size [17]. In this study, current smokers had higher LDL-C and TG levels as well as lower HDL levels than never smokers. Former smokers also had higher TG levels than never smokers. The frequency of EC smoking has increased, particularly owing to the transition from TC smoking to reduce exposure to harmful contents of TCs or bridge the cessation of TC

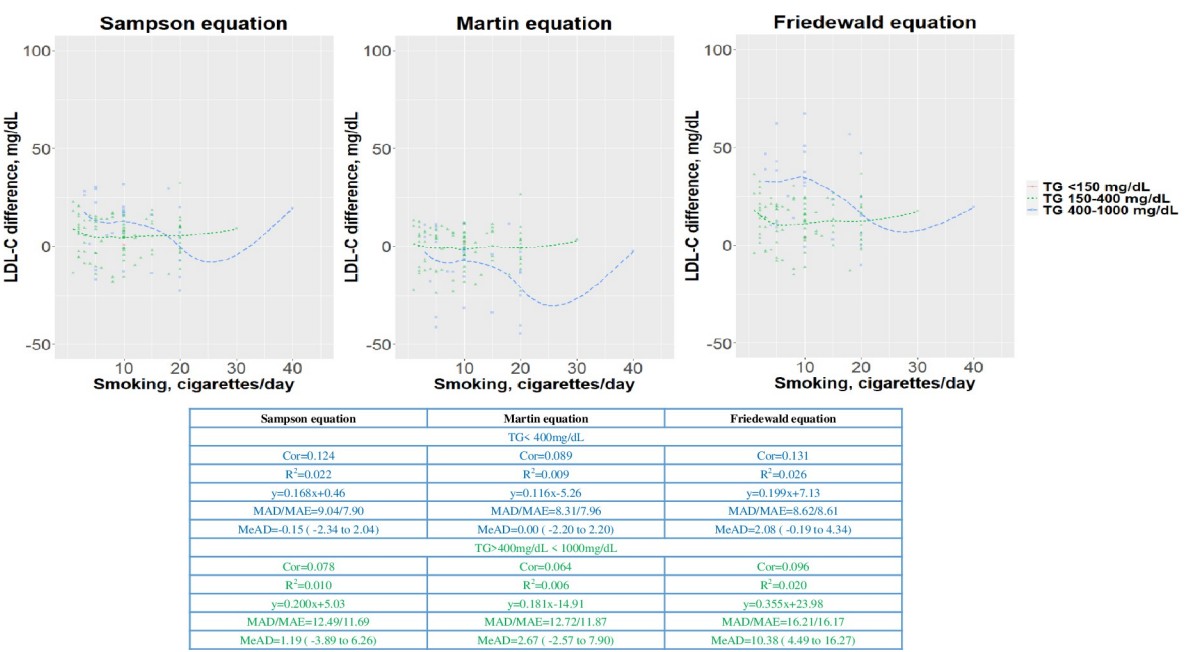

| Sampson equation | Martin equation | Friedewald equation |
|---|---|---|
| TG< 400mg/dL | | |
| Cor=0.124 | Cor=0.089 | Cor=0.131 |
| R²=0.022 | R²=0.009 | R²=0.026 |
| y=0.168x+0.46 | y=0.116x-5.26 | y=0.199x+7.13 |
| MAD/MAE=9.04/7.90 | MAD/MAE=8.31/7.96 | MAD/MAE=8.62/8.61 |
| MeAD=-0.15 ( -2.34 to 2.04) | MeAD=0.00 ( -2.20 to 2.20) | MeAD=2.08 ( -0.19 to 4.34) |
| TG>400mg/dL < 1000mg/dL | | |
| Cor=0.078 | Cor=0.064 | Cor=0.096 |
| R²=0.010 | R²=0.006 | R²=0.020 |
| y=0.200x+5.03 | y=0.181x-14.91 | y=0.355x+23.98 |
| MAD/MAE=12.49/11.69 | MAD/MAE=12.72/11.87 | MAD/MAE=16.21/16.17 |
| MeAD=1.19 ( -3.89 to 6.26) | MeAD=2.67 ( -2.57 to 7.90) | MeAD=10.38 ( 4.49 to 16.27) |

**Fig 4. Residual error plots for low-density lipoprotein cholesterol (LDL-C) level derived by equations according to daily electronic cigarette smoking.** Differences between the estimates obtained by the Sampson equation (A), Martin equation (B), and Friedewald equation and dLDL-C, stratified by daily electronic cigarette smoking. The residual error is calculated by the difference between direct LDL-C (dLDL-C), and the values are obtained using the lipid equations. The dots indicate the individual samples, colored according to smoking status. The color scale indicates individuals. The solid line indicates the trend by the local regression method. The median absolute deviations with 95% confidence intervals were calculated by two-sample difference. Cor, correlation coefficient; MAD, mean absolute deviation; MAE, mean absolute error; MeAD, median absolute deviation; R2, correlation coefficient.

smoking. In this study, EC was common among young and middle-aged adults. Limited data are available on the effects of ECs.

## Differences in LDL-C and dLDL-C depending on the equation

Higher differences in LDL-C and dLDL-C levels (in terms of the MADs and MAEs) were observed in current smokers than in never and former smokers, while slightly lower differences were observed in never and former smokers. The MADs, MAEs, residual errors, and 95% CIs of MeADs obtained with the Martin equation in never, former, and current smokers were lower than those obtained with the other equations. The errors remained high for LDL levels <40 mg/dL in all formulas, suggesting limitations in their practical application. In the current smoker group, the MADs and MeADs for TG levels <150 mg/dL were lower than those in the never smoker group. For TG levels <150 mg/dL, the Martin equation yielded lower MADs and MeADs than the other equations, except for those with LDL levels <40 mg/dL in the current smoker group.

## Differences in LDL-C and dLDL-C in current smokers

Direct LDL-C measurement is the best approach for accurate measurement of lipid profiles; however, direct measurement is not available in primary clinics and is not cost-effective. Indirect measurements are used in clinical practice for various reasons; therefore, a precise method for indirect calculation of LDL-C is needed. The three well-known equations used in this study exhibited inaccuracies at TG levels >400 mg/dL; the Friedewald equation exhibited a higher positive error than the other equations, while the Martin equation exhibited a slightly higher

positive error. Furthermore, the Sampson method exhibited a higher likelihood of underestimating LDL-C levels than the Martin equation at TG levels <150 mg/dL. Moreover, achieving an LDL-C level <40 mg/dL in patients with a positive smoking status led to more precise MAD values when using the Martin equation. The Sampson equation demonstrated greater precision in patients for TG levels >400 mg/dL; however, its application is recommended for patients with low LDL levels. In case of LDL-C levels <70 mg/dL and TG levels >400 mg/dL, the MAD of the Sampson equation was lower than those of the other equations without never smokers. In case of LDL-C levels <70 mg/dL and TG levels > 400 mg/dL with never smokers, the Martin equation had a lower MAD than the other equations.

## Limitations

This study has certain limitations. First, dLDL-C levels were measured using an enzymatic method, whereas beta quantification (ultracentrifugation) was conducted in most studies. This method is regarded as the standard measurement technique for LDL-C; however, the lipid data from the KNHANES underwent validation through a standardization program from the United States Centers for Disease Control and Prevention. Second, genetic components are associated with lipid metabolism and lipid profiles; however, these factors could not be explored owing to the absence of genetic data. Third, metabolic components, insulin resistance, and effects of statins were not evaluated. Fourth, this cross-sectional study did not include follow-up measurements of blood samples; therefore, we were unable to assess follow-up data. Fifth, the sample sizes of the high TG level (>500 mg/dL) and low LDL-C level (<40 mg/dL) groups were small. Sixth, the levels of small lipid components such as VLDL, apoprotein A/B, and lipoprotein A were not measured. Further studies are needed with large sample sizes and data on VLDL-cholesterol, apoprotein, and measured lipoprotein A.

## Conclusions

This study utilized a nationally representative sample and validated the precision of LDL-C estimation in patients with smoking habits. Our investigation revealed variations in the precision of estimated LDL-C levels according to smoking status. In those with TG levels <400 mg/dL, daily smoking of ECs or TCs did not have a significant impact on LDL estimations. We recommend the application of the Sampson equation for those with TG levels >400 mg/dL and the Martin equation for those with TG levels <400 mg/dL, regardless of smoking status. Using different equations may help minimize the errors in LDL-C measurements.

## Supporting information

**S1 Fig. Flow diagram of the inclusion and exclusion criteria for participants from the KNHANES (2009–2021).** KNHANES, Korea National Health and Nutrition Examination Survey; LDL-C, low-density lipoprotein cholesterol; dLDL-C, direct low-density lipoprotein cholesterol.
(DOCX)

**S2 Fig. Correlation of estimated vs. direct low-density lipoprotein cholesterol (LDL-C) levels in patients with smoking status.** A. The Sampson equation vs. direct LDL-C for never smokers. B. The Martin equation vs. direct LDL-C for never smokers. C. The Friedewald equation vs. direct LDL-C for never smokers. D. The Sampson equation vs. direct LDL-C for former smokers. E. The Martin equation vs. direct LDL-C for former smokers. F. The Friedewald equation vs. direct LDL-C for former smokers. G. The Sampson equation vs. direct LDL-C for current smokers. H. The Martin equation vs. direct LDL-C for current smokers. I. The

Friedewald equation vs. direct LDL-C for current smokers. J. The Sampson equation vs. direct LDL-C for electronic cigarette smokers. K. The Martin equation vs. direct LDL-C for electronic cigarette smokers. L. The Friedewald equation vs. direct LDL-C for electronic cigarette smokers. Cor, correlation coefficient; MAE, mean absolute error; $R^2$, correlation coefficient; RMSE, root mean square error; TG, triglyceride. The dots indicate the individual samples colored according to TG level. The color scale indicates individuals. SI conversion factors: To convert cholesterol to mmol/L, values were multiplied by 0.0259.
(DOCX)

**S3 Fig. Mean absolute deviations of directly measured low-density lipoprotein cholesterol (dLDL-C) stratified by estimated LDL-C.** (A) MAD of estimated LDL-C vs. dLDL-C for the never smoker group (TG levels <1000 mg/dL). (B) MAD of estimated LDL-C vs. dLDL-C for the never smoker group (TG levels <150 mg/dL). (C) MAD of estimated LDL-C vs. dLDL-C for the former smoker group (TG levels <1000 mg/dL). (D) MAD of estimated LDL-C vs. dLDL-C for the former smoker group (TG levels <150 mg/dL). (E) MAD of estimated LDL-C vs. dLDL-C for the current smoker group (TG levels <150 mg/dL). (F) MAD of estimated LDL-C vs. dLDL-C for the current smoker group (TG levels <150 mg/dL). (G) MAD of estimated LDL-C vs. dLDL-C for the EC smoker group (TG levels <150 mg/dL). (H) MAD of estimated LDL-C vs. dLDL-C for the EC smoker group (TG levels <150 mg/dL). HDL, high-density lipoprotein; TG, triglyceride; EC, electronic cigarette. Sampson equation (blue), Martin equation (red), and Friedewald equation (green).
(DOCX)

**S4 Fig. Mean absolute deviations of directly measured low-density lipoprotein cholesterol (dLDL-C) stratified by non-high-density lipoprotein (NHDL).** (A) MAD of estimated LDL-C vs. dLDL-C for the never smoker group (TG levels <1000 mg/dL). (B) MAD of estimated LDL-C vs. dLDL-C for the former smoker group (TG levels <1000 mg/dL). (C) MAD of estimated LDL-C vs. dLDL-C for the current smoker group (TG levels <150 mg/dL). (D) MAD of estimated LDL-C vs. dLDL-C for the EC smoker group (TG levels <150 mg/dL. NHDL, non-high-density lipoprotein; TG, triglyceride; EC, electronic cigarette. Sampson equation (blue), Martin equation (red), and Friedewald equation (green).
(DOCX)

**S5 Fig. Residual error plots for low-density lipoprotein cholesterol (LDL-C) level estimated using equations according to smoking status (triglyceride levels <1000 mg/dL).** Differences between the estimates obtained by the Sampson equation (A), Martin equation (B), and Friedewald equation (C) and direct LDL-C (dLDL-C), stratified by triglyceride (TG) level. Residual error was calculated by the difference between direct LDL-C (dLDL-C) and the values obtained using the equations. The dots indicate the individual samples, colored according to smoking status. The color scale indicates individuals. The solid line indicates the trend by the local regression method. Cor, correlation coefficient; MAD, mean absolute deviation; MAE, mean absolute error; R2 indicates correlation coefficient.
(DOCX)

**S1 Table. Characteristics of study population with triglyceride levels of ≥400 mg/dL and <1000 mg/dL.**
(DOCX)

**S2 Table. Characteristics of the study population with triglyceride levels of ≥150 mg/dL and <400 mg/dL.**
(DOCX)

**S3 Table. Characteristics of the study population with triglyceride levels of <150 mg/dL.**
(DOCX)

**S4 Table. Mean and median absolute deviations with 95% confidence intervals of estimated low-density lipoprotein cholesterol stratified by dLDL-C in the group with TG levels of <400 mg/dL.**
(DOCX)

**S5 Table. Mean and median absolute deviations with 95% confidence intervals of estimated low-density lipoprotein cholesterol stratified by dLDL-C in the group with TG levels of <1000 mg/dL.**
(DOCX)

**S6 Table. Mean and median absolute deviations with 95% confidence intervals of estimated low-density lipoprotein cholesterol stratified by dLDL-C in the group with TG levels of >400 mg/dL and <1000 mg/dL.**
(DOCX)

## Acknowledgments

We are grateful to the Korean Ministry of Health and Welfare for providing the data.

## Author Contributions

**Conceptualization:** Han-Joon Bae, Seung-Pyo Hong.

**Data curation:** Han-Joon Bae, Hae Won Jung.

**Formal analysis:** Han-Joon Bae, Hae Won Jung.

**Funding acquisition:** Han-Joon Bae.

**Investigation:** Han-Joon Bae, Hae Won Jung.

**Methodology:** Hae Won Jung, Seung-Pyo Hong.

**Project administration:** Han-Joon Bae, Seung-Pyo Hong.

**Resources:** Han-Joon Bae, Hae Won Jung, Seung-Pyo Hong.

**Software:** Han-Joon Bae, Hae Won Jung.

**Supervision:** Seung-Pyo Hong.

**Validation:** Han-Joon Bae.

**Visualization:** Han-Joon Bae, Hae Won Jung.

**Writing – original draft:** Han-Joon Bae.

**Writing – review & editing:** Han-Joon Bae, Seung-Pyo Hong.

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
