## [Decision Letter · Decision Letter 0]

5 Jun 2024

PONE-D-24-08009More precise method of low-density lipoprotein cholesterol estimation for tobacco and electronic cigarette smokers: a cross-sectional studyPLOS ONE

Dear Dr. Bae,

Thank you for submitting your manuscript to PLOS ONE. After careful consideration, we feel that it has merit but does not fully meet PLOS ONE’s publication criteria as it currently stands. Therefore, we invite you to submit a revised version of the manuscript that addresses the points raised during the review process.

We look forward to receiving your revised manuscript.

Kind regards,

Shukri AlSaif

Academic Editor

PLOS ONE

Journal Requirements:

Reviewers' comments:

Reviewer's Responses to Questions

**Comments to the Author**

1. Is the manuscript technically sound, and do the data support the conclusions?

Reviewer #1: Yes

Reviewer #2: Partly

2. Has the statistical analysis been performed appropriately and rigorously? 

Reviewer #1: Yes

Reviewer #2: Yes

3. Have the authors made all data underlying the findings in their manuscript fully available?

Reviewer #1: Yes

Reviewer #2: Yes

4. Is the manuscript presented in an intelligible fashion and written in standard English?

Reviewer #1: Yes

Reviewer #2: Yes

5. Review Comments to the Author

Reviewer #1: This work investigated the accuracy of three equations used for LDL-C calculation in smokers. Several points should be addressed before publication.

1. The rationale for this analysis should be elaborated more. Is calculated LDL-C really needed? Why don't you just measure LDL-C directly? If calculated LDL-C is needed, what is the rationale? Whether it is availability problem, technical difficulty of the test and/or cost problem, I think it should be described in the Introduction.

2. Why did you decide that analysis was done according to the smoking status? Was calculated LDL-C less accurate in smokers compared to non-smokers?

3. Related to #2, you described the relationship between smoking and lipid profile in the Discussion section (line 262-277). I think this may be moved to the Introduction, because it helps to justify why the analysis should be done according to the smoking status.

4. What is the novelty of the work? It seems that the Conclusion just confirms that the study finding is consistent with the pre-existing knowledge overall.

Reviewer #2: 1. The tables S1, S2, and S3 should include posthoc test results for significant comparisons. Additionally, significant results should be highlighted in bold to enhance the readability and comprehension of the tables.

2. Given that the comparisons are made within the same sample, adjusted p-values (e.g., using the Benjamini-Hochberg method) should be provided to control for multiple testing.

3. The statement in line 110 should indicate that TG <150 is excluded.

4. The font sizes in the axes and titles of all figures should be increased to improve readability.

5. The definition of overall precision as the ratio of direct LDL-C (dLDL-C) to non-high-density lipoprotein (non-HDL-C) should be supported by a reference or a clear rationale, such as citing relevant studies or providing a logical explanation for why this ratio is an appropriate measure

6. In Figure 2, the LDL-C ranges should be explicitly labeled on the plots for better clarity.

7. Figure 2 currently presents TG <150 mg/dL and TG <1000 mg/dL. It should categorize as TG ≥150–TG <400 mg/dL and TG ≥400–TG <1000 mg/dL. Including TG <150 mg/dL in the <1000 mg/dL category can lead to complexity in comparisons.

8. In line 212, the comment on TG categories is confusing. Table 1 presents data for TG <150 mg/dL, Table S4 for <400 mg/dL, and Table S5 for <1000 mg/dL. However, the text mentions TG levels <150 mg/dL, 150–400 mg/dL, and 400–1000 mg/dL. The TG categories should be clarified consistently throughout the manuscript.

9. The issue of inconsistent TG categories appears in multiple places. The TG categories should be clearly defined and consistently used throughout the manuscript.

10. In Figure 3, the legend should be reordered to present TG <150, 150–400 mg/dL, and 400–1000 mg/dL sequentially to improve the clarity of the plot.

11. The statement in the results about the superiority of the Martin equation in Figure 3 should be moderated. The results suggest that the equations perform similarly.

12. The findings related to Figure 4 should be included in the manuscript to provide a complete interpretation of the results.

13. The performance of the equations in the category of LDL-C <70 mg/dL and TG >400 mg/dL should be evaluated. These results could provide valuable insights into the literature.

14. A more detailed discussion comparing the results with existing literature should be provided. Comparing these findings with studies conducted in different populations would help to understand the consistency and differences in the results.

6. PLOS authors have the option to publish the peer review history of their article (what does this mean?). If published, this will include your full peer review and any attached files.

Reviewer #1: No

Reviewer #2: No

---

## [Author Response · Author response to Decision Letter 0]

17 Jul 2024

Thank you for your thoughtful and thorough review of our manuscript. We have made every effort to fulfil the reviewers’ specific requests. We have provided a point-by-point response to all comments from the editorial committee and reviewers below and also indicated our revisions to the manuscript in response to each comment.

Review Comments to the Author

Reviewer #1: This work investigated the accuracy of three equations used for LDL-C calculation in smokers. Several points should be addressed before publication.

1. The rationale for this analysis should be elaborated more. Is calculated LDL-C really needed? Why don't you just measure LDL-C directly? If calculated LDL-C is needed, what is the rationale? Whether it is availability problem, technical difficulty of the test and/or cost problem, I think it should be described in the Introduction.

▶Response: We appreciate the reviewers’ comments. Accordingly, the following text has been added to the Introduction (3rd paragraph): 

“The best method for measuring LDL-C levels uses ultracentrifuged plasma, but this method is time-consuming and labor-intensive [11]. Therefore, several homogeneous direct methods to measure high-density lipoprotein cholesterol (HDL-C) and LDL-C levels have been developed. These methods use various surfactants, ionic polymers, and other components to measure cholesterol levels in specific classes of lipoproteins in the serum [12]; however, they can be expensive and are not available at all medical centers.”

2. Why did you decide that analysis was done according to the smoking status? Was calculated LDL-C less accurate in smokers compared to non-smokers?

▶Response: Thank you for your comment. We have added the following information to the Introduction (second paragraph) to clarify this aspect: 

“…, and both smoking and smoking cessation have been shown to be associated with changes in cholesterol levels [9, 10].”

3. Related to #2, you described the relationship between smoking and lipid profile in the Discussion section (line 262-277). I think this may be moved to the Introduction, because it helps to justify why the analysis should be done according to the smoking status.

▶Response: Thank you for your comment, which we agree with. As mentioned in the response to the previous comment, we have now referred to this aspect in the Introduction (third paragraph).

“and both smoking and smoking cessation have been shown to be associated with changes in cholesterol levels [9, 10].”

4. What is the novelty of the work? It seems that the Conclusion just confirms that the study finding is consistent with the pre-existing knowledge overall.

▶Response: Thank you for pointing this out. We agree with your comment, and research on LDL validation has indeed been conducted in several countries and research institutes. However, research on validation according to smoking status, particularly for users of electronic cigarettes, is lacking. Accordingly, we have added the following text to the Introduction (third paragraph):

“Moreover, the use of electronic cigarettes (ECs) has increased considerably recently, but there have been very few studies on LDL-C measurements in this context.” 

Reviewer #2: 

1. The tables S1, S2, and S3 should include posthoc test results for significant comparisons. Additionally, significant results should be highlighted in bold to enhance the readability and comprehension of the tables.

▶Response: Thank you for your helpful comments. Accordingly, we have highlighted the significant results in bold in Tables S1, S2, and S3, and added adjusted p-values derived using the Benjamini–Hochberg post-hoc method to the tables.

2. Given that the comparisons are made within the same sample, adjusted p-values (e.g., using the Benjamini-Hochberg method) should be provided to control for multiple testing.

▶Response: Thank you for pointing this out. As mentioned in the response to the previous comment, the required changes have now been made to Tables S1, S2, and S3.

S1 Table. Characteristics of the study population with triglyceride levels of ≥400 mg/dL and <1000 mg/dL 

Smoke status

Adjusted p-values (Benjamini–Hochberg method) Never Never Never Former Former Current 

 Former Current Electronic Current Electronic Electronic 

Age, years 0.037 0.001 <0.001 <0.001 <0.001 0.001

Male, % <0.001 <0.001 <0.001 0.101 0.658 0.244

Body mass index, kg/m2 0.602 0.602 0.768 0.602 0.602 0.602

Laboratory data 　 　 　 　 　 

 Fasting glucose, mg/dL 0.088 0.600 0.108 0.137 0.030 0.088

 Blood urea nitrogen, mg/dL <0.001 0.691 0.691 <0.001 0.016 0.691

 Creatinine, mg/dL <0.001 <0.001 <0.001 0.002 0.350 0.523

 GFR_EPI 0.004 0.224 0.168 <0.001 <0.001 0.224

Cholesterol 　 　 　 　 　 

 Total cholesterol, mg/dL 0.998 0.998 0.998 0.998 0.998 0.998

 Triglyceride, mg/dL 0.808 <0.001 0.492 0.006 0.530 0.492

 HDL-C, mg/dL 0.971 0.971 0.971 0.971 0.971 0.971

 Direct LDL-C, mg/dL 0.983 0.642 0.144 0.642 0.144 0.158

 Non-HDL-C, mg/dL 0.978 0.978 0.978 0.978 0.978 0.978

 Sampson, mg/dL 0.416 0.054 0.977 0.416 0.690 0.416

 Martin, mg/dL 0.387 0.066 0.917 0.387 0.661 0.387

 Friedewald, mg/dL 0.366 0.018 0.902 0.366 0.707 0.366

Positive absolute value Martin, mg/dL 0.877 0.006 0.030 0.006 0.024 0.364

Positive absolute value Sampson, mg/dL 0.217 <0.001 0.080 0.003 0.204 0.993

Positive absolute value Friedewald, mg/dL 0.181 <0.001 0.056 <0.001 0.181 0.740

S2 Table. Characteristics of the study population with triglyceride levels of ≥150 mg/dL and <400 mg/dL

Smoke status

adjusted p-values (Benjamini–Hochberg method) Never Never Never Former Former Current 

 Former Current Electronic Current Electronic Electronic

Age, years 0.001 <0.001 <0.001 <0.001 <0.001 <0.001

Male, % <0.001 <0.001 <0.001 0.021 0.901 0.5436

Body mass index, kg/m2 0.048 0.888 0.174 0.066 0.634 0.174

Laboratory data 　 　 　 　 　 　

 Fasting glucose, mg/dL <0.001 0.020 0.413 <0.001 <0.001 0.084

 Blood urea nitrogen, mg/dL <0.001 0.517 0.396 <0.001 <0.001 0.478

 Creatinine, mg/dL <0.001 <0.001 <0.001 <0.001 0.014 0.853

 GFR_EPI <0.001 0.018 0.035 <0.001 <0.001 <0.001

Cholesterol 　 　 　 　 　 　

 Total cholesterol, mg/dL 0.006 0.201 0.709 0.201 0.201 0.378

 Triglyceride, mg/dL <0.001 <0.001 <0.001 0.008 0.030 0.338

 HDL-C, mg/dL 0.042 <0.001 0.219 0.219 0.719 0.719

 Direct LDL-C, mg/dL 0.012 0.329 0.195 0.006 0.018 0.329

 Non-HDL-C, mg/dL 0.024 0.547 0.518 0.100 0.100 0.500

 Sampson, mg/dL <0.001 0.009 0.681 0.378 0.378 0.538

 Martin, mg/dL <0.001 0.090 0.934 0.224 0.224 0.541

 Friedewald, mg/dL <0.001 0.003 0.541 0.461 0.461 0.541

Positive absolute value Martin, mg/dL 0.605 <0.001 0.003 <0.001 0.002 0.149

Positive absolute value Sampson, mg/dL 0.003 <0.001 0.968 0.581 0.248 0.140

Positive absolute value Friedewald, mg/dL <0.001 <0.001 <0.001 <0.001 0.004 0.810

S3 Table. Characteristics of the study population with triglyceride levels of <150 mg/dL 

Smoke status

Adjusted p-values (Benjamini–Hochberg method) Never Never Never Former Former Current 

 Former Current Electronic Current Electronic Electronic 

Age, years <0.001 <0.001 0.026 <0.001 <0.001 <0.001

Male, % <0.001 <0.001 <0.001 0.2964 0.2385 0.49

Body mass index, kg/m2 <0.001 0.008 0.074 <0.001 0.253 0.454

Laboratory data 　 　 　 　 　 　

 Fasting glucose, mg/dL <0.001 0.021 0.302 <0.001 0.012 0.959

 Blood urea nitrogen, mg/dL <0.001 <0.001 0.475 <0.001 <0.001 0.014

 Creatinine, mg/dL <0.001 <0.001 <0.001 <0.001 0.024 0.845

 GFR_EPI <0.001 <0.001 0.700 <0.001 <0.001 <0.001

Cholesterol 　 　 　 　 　 　

 Total cholesterol, mg/dL 0.904 0.369 0.904 0.369 0.904 0.904

 Triglyceride, mg/dL <0.001 <0.001 0.004 0.611 0.857 0.857

 HDL-C, mg/dL <0.001 <0.001 0.044 0.410 0.730 0.882

 Direct LDL-C, mg/dL 0.555 0.779 0.863 0.546 0.906 0.779

 Non-HDL-C, mg/dL 0.036 0.843 0.835 0.099 0.835 0.835

 Sampson, mg/dL 0.297 0.340 0.886 0.120 0.678 0.784

 Martin, mg/dL 0.159 0.548 0.972 0.126 0.692 0.797

 Friedewald, mg/dL 0.282 0.282 0.818 0.096 0.656 0.784

Positive absolute value Martin, mg/dL 0.552 0.030 0.500 0.234 0.552 0.887

Positive absolute value Sampson, mg/dL 0.597 0.363 0.402 0.402 0.402 0.363

Positive absolute value Friedewald, mg/dL 0.600 0.600 0.805 0.988 0.600 0.600

3. The statement in line 110 should indicate that TG <150 is excluded.

▶Response: Thank you for pointing this out. Accordingly, the corresponding text has been modified as follows: 

“At each TG level, the current smoker group exhibited higher TG levels (except those with TG levels <150 mg/dL) and dLDL-C levels (except those with TG levels <150 mg/dL) than the never smoker group (except those with TG levels <150 mg/dL).”

4. The font sizes in the axes and titles of all figures should be increased to improve readability.

▶Response: Thank you for this helpful comment. Accordingly, we have increased the font sizes of all axes and titles in the figures.

5. The definition of overall precision as the ratio of direct LDL-C (dLDL-C) to non-high-density lipoprotein (non-HDL-C) should be supported by a reference or a clear rationale, such as citing relevant studies or providing a logical explanation for why this ratio is an appropriate measure

▶Response: Thank you for your insightful comments. 

Non-HDL-C, calculated by subtracting HDL-C from total cholesterol, includes all particles that cause cardiovascular disease, namely, LDL-C, very low-density lipoprotein (VLDL), intermediate density lipoprotein, and lipoprotein A cholesterol. Non-HDL-C does not distinguish between LDL-C and remnant cholesterol and is considered a comprehensive target for managing cardiovascular risk. Clinically, LDL-C levels are related to non-HDL-C values in most cases; however, some patients have significantly higher non-HDL-C levels than LDL-C levels. 

1. Nordestgaard B.G. Quantifying atherogenic lipoproteins for lipid-lowering strategies: consensus-based recommendations from EAS and EFLM. Atherosclerosis. 2020; 294: 46-61

2. Nordestgaard B.G. Triglyceride-rich lipoproteins and atherosclerotic cardiovascular disease: new insights from epidemiology, genetics, and biology. Circ Res. 2016; 118: 547-563

Accordingly, the corresponding text has been modified as follows: Method statistical analyses 1st pararagh

“Overall precision was defined as the ratio of direct LDL-C (dLDL-C) to estimated LDL-C and non-high-density lipoprotein (NHDL). (S2 Fig, S3 Fig and S4 Fig).”

6. In Figure 2, the LDL-C ranges should be explicitly labeled on the plots for better clarity.

▶Response: Thank you for pointing this out. We have now added labels for the LDL-C ranges and the axis indexes.

7. Figure 2 currently presents TG <150 mg/dL and TG <1000 mg/dL. It should categorize as TG ≥150–TG <400 mg/dL and TG ≥400–TG <1000 mg/dL. Including TG <150 mg/dL in the <1000 mg/dL category can lead to complexity in comparisons.

▶Response: Thank you for your insightful comment. To illustrate the LDL-C error ranges for TG <150 mg/dL and the more commonly used TG <1000 mg/dL, Figure 2 depicts the TG <150 and TG <1000 mg/dL ranges. Additional data for the TG ≥150 and <400 mg/dL and TG ≥400 and <1000 mg/dL groups have now been included in the supplementary material.

8. In line 212, the comment on TG categories is confusing. Table 1 presents data for TG <150 mg/dL, Table S4 for <400 mg/dL, and Table S5 for <1000 mg/dL. However, the text mentions TG levels <150 mg/dL, 150–400 mg/dL, and 400–1000 mg/dL. The TG categories should be clarified consistently throughout the manuscript.

▶Response: Thank you for your comment. In real-world applications, we divided the groups to show the error when TG levels are below 400 mg/dL and TG levels are 1000 mg/dL. In the real world, we classified the difference between TGs below 400 and 1000 differently from the existing categories to explain the variation by LDL level. It has been modified like this.

The Friedewald equation demonstrated higher MADs and MeADs than the other equations for TG levels <150 mg/dL, <400 mg/dL, <1000 mg/d, and 400-1000 mg/dL according to smoking status (Tables 1, S4,S5 and S6).

9. The issue of inconsistent TG categories appears in multiple places. The TG categories should be clearly defined and consistently used throughout the manuscript.

▶Response: Thank you for your comment. As mentioned in the response to the previous comment, the relevant data has been added and changes have now been made to the text.

10. In Figure 3, the legend should be reordered to present TG <150, 150–400 mg/dL, and 400–1000 mg/dL sequentially to improve the clarity of the plot.

▶Response: Thank you for your comment. We have changed the sequence of the TG groups as follows: 

11. The statement in the results about the superiority of the Martin equation in Figure 3 should be moderated. The results suggest that the equations perform similarly.

▶Response: Thank you for your comment. Accordingly, we have changed the text as follows (Results - Precision of estimated LDL-C for the group with daily smoker status):

“The accuracies of the Martin and Sampson equations were similar to that of the Friedewald equation.”

12. The findings related to Figure 4 should be included in the manuscript to provide a complete interpretation of the results.

▶Response: Thank you for your comment. Accordingly, we have added the following text to the manuscript (Results - Precision of estimated LDL-C for the group with daily smoker status):

“In the EC smoker group with TG levels <400 mg/dL and >400 mg/dL, the MADs, MeADs, and correlation coefficients for the Sampson and Martin equations were similar. The Friedewald equation exhibited a higher difference than the other equations.”

13. The performance of the equations in the category of LDL-C <70 mg/dL and TG >400 mg/dL should be evaluated. These results could provide valuable insights into the literature.

▶Response: Thank you for your valuable suggestions. Accordingly, we have provided the corresponding data in S6 Table and added the following text to the Discussion (Differences in LDL-C and dLDL-C in current smokers): 

“In case of LDL-C levels <70 mg/dL and TG levels >400 mg/dL, the MAD of the Sampson equation was lower than those of the other equations without never smokers. In case of LDL-C levels <70 mg/dL and TG levels > 400 mg/dL with never smokers, the Martin equation had a lower MAD than the other equations.” 

S6 Table. Mean and median absolute deviations with 95% confidence intervals of estimated low-density lipoprotein cholesterol stratified by dLDL-C in the group with TG levels >400 mg/dL and <1000 mg/dL

Sampson equation Martin equation Friedewald equation

MAD MeAD MAD MeAD MAD MeAD

 Never smoker 

9.81 -2.52 (-11.16 to 6.12) 9.49 -1.78 (-10.29 to 6.74) 14.11 -10.06 (-20.09 to -0.04)

 Former smoker 

9.91 -5.34 (-9.82 to -0.85) 10.36 -7.12 (-11.96 to -2.27) 16.99 -15.21 (-21.71 to -8.72)

 Current smoker 

11.87 -9.34 (-13.88 to -4.80) 12.45 -7.56 (-11.66 to -3.47) 20.23 -17.49 (-23.57 to -11.42)

 Electronic smoker 

12.83 -15.12 (-39.4 to 9.16) 13.00 -13.71 (-37.40 to 9.97) 17.05 -19.45 (-50.38 to 11.47)

14. A more detailed discussion comparing the results with existing literature should be provided. Comparing these findings with studies conducted in different populations would help to understand the consistency and differences in the results

▶Response: Thank you for your comment. However, although numerous reports on LDL validation have been published recently, it is difficult to make comparisons due to insufficient literature on validation according to smoking status. In fact, the lack of data in this context was the main motivation behind designing and conducting this study.

---

## [Decision Letter · Decision Letter 1]

5 Aug 2024

More precise method of low-density lipoprotein cholesterol estimation for tobacco and electronic cigarette smokers: a cross-sectional study

PONE-D-24-08009R1

Dear Dr. Bae,

We’re pleased to inform you that your manuscript has been judged scientifically suitable for publication and will be formally accepted for publication once it meets all outstanding technical requirements.

Kind regards,

Shukri AlSaif

Academic Editor

PLOS ONE

Review Comments to the Author

Reviewer #1: Comments from the reviewer has been properly addressed and the manuscript has been improved.

Thanks for your effort.

Reviewer #2: I would like to express my gratitude to the authors for their attention to and implementation of the suggested corrections.

---

## [Editor Report · Acceptance letter]

12 Sep 2024

PONE-D-24-08009R1 

PLOS ONE

Dear Dr. Bae, 

I'm pleased to inform you that your manuscript has been deemed suitable for publication in PLOS ONE. Congratulations! Your manuscript is now being handed over to our production team.

Kind regards, 

on behalf of

Dr. Shukri AlSaif 

Academic Editor

PLOS ONE